ecology/oceanography/behaviour

passive acoustic monitoring, humpback whale, *Megaptera novaeangliae*, feeding ground, Southern Ocean, Weddell Sea

**Author for correspondence:**
Elena Schall
e-mail: elena.schall@awi.de

# Large-scale spatial variabilities in the humpback whale acoustic presence in the Atlantic sector of the Southern Ocean

Elena Schall[1], Karolin Thomisch[1], Olaf Boebel[1], Gabriele Gerlach[2,3], Stefanie Spiesecke[1] and Ilse Van Opzeeland[1,2]

[1]Ocean Acoustics Lab, Alfred Wegener Institute for Polar and Marine Research, Klußmannstraße 3d, 27570 Bremerhaven, Germany
[2]Helmholtz Institute for Functional Marine Biodiversity, Carl von Ossietzky University Oldenburg, Ammerländer Heerstraße 231, 26129 Oldenburg, Germany
[3]Animal Biodiversity and Evolutionary Biology, Carl von Ossietzky University Oldenburg, Ammerländer Heerstraße 114-118, 26129 Oldenburg, Germany

ES, 0000-0002-7740-5466

Southern Hemisphere humpback whales (*Megaptera novaeangliae*) inhabit a wide variety of ecosystems including both low- and high-latitude areas. Understanding the habitat selection of humpback whale populations is key for humpback whale stock management and general ecosystem management. In the Atlantic sector of the Southern Ocean (*ASSO*), the investigation of baleen whale distribution by sighting surveys is temporally restricted to the austral summer. The implementation of autonomous passive acoustic monitoring, in turn, allows the study of vocal baleen whales year-round. This study describes the results of analysing passive acoustic data spanning 12 recording positions throughout the *ASSO* applying a combination of automatic and manual analysis methods to register humpback whale acoustic activity. Humpback whales were present at nine recording positions with higher acoustic activities towards lower latitudes and the eastern and western edges of the *ASSO*. During all months, except December (the month with the fewest recordings), humpback whale acoustic activity was registered in the *ASSO*. The acoustic presence of humpback whales at various locations in the *ASSO* confirms previous observations that part of the population remains in high-latitude waters beyond austral summer, presumably to feed. The spatial and temporal extent

of humpback whale presence in the *ASSO* suggests that this area may be used by multiple humpback whale breeding populations as a feeding ground.

## 1. Introduction

Humpback whales (*Megaptera novaeangliae*) inhabit all major oceans and have adapted to diverse ecosystems, including polar and subpolar ecosystems mainly to feed during the summer months, and equatorial ecosystems almost exclusively to breed and calve throughout the winter months (e.g. [1–5]). To reach the most productive feeding areas, humpback whales undertake one of the longest mammalian migrations, stretching between their low-latitude breeding grounds and mid- to high-latitude feeding grounds [6,7]. As in other baleen whale species, migratory behaviour, in humpback whales, is characterized by population-specific spatio-temporal patterns, but is also flexible in terms of destinations and timing, including the omission or delay of migration or the spatial adaptation of migration routes [3,7–10]. Less extreme migratory deviations are very common in many baleen whale populations worldwide. Individuals or groups of baleen whales frequently extend their stay in productive feeding areas beyond the summer months in order to maximize energy uptake [9,11]. The Southern Ocean includes the most important feeding areas for baleen whales in the Southern Hemisphere [12], but knowledge on the year-round distribution of baleen whales in many regions of the Southern Ocean is still limited due to the restricted accessibility of these areas outside the summer months. Baseline information on baleen whale distribution and ecology is key for understanding their role as large predators in structuring the Southern Ocean ecosystem [13].

One presumed high-latitude feeding area for humpback whales is the Atlantic sector of the Southern Ocean (hereinafter referred to as *ASSO*). The *ASSO* is equivalent to the management area II defined by the International Whaling Commission (IWC) and is thought to serve as a feeding area for two humpback whale breeding stocks from the South Atlantic: breeding stock A from the southwest Atlantic and breeding stock B from the southeast Atlantic [14]. The *ASSO* is a typical Southern Ocean ecosystem dominated by sea ice dynamics, the Antarctic Circumpolar Current (ACC) and associated fronts and boundaries, and the Weddell Sea Gyre [15–17]. Sea ice concentration and extent both during winter and summer have major effects on primary as well as secondary production [15]. The Southern Boundary of the ACC creates various productivity hotspots around the Antarctic continent due to its high concentration of nutrient-rich Upper Circumpolar Deep Water [18]. The Weddell Gyre acts as an insulating current system which regulates temperature in the Weddell Sea and efficiently circulates nutrients, phytoplankton and zooplankton throughout great parts of the *ASSO* [16,19]. Both the Weddell Gyre and the ACC function as transport mechanisms (e.g. the 'conveyor belt') for the recruitment of zooplankton larvae from other Antarctic regions such as the West Antarctic Peninsula [19]. In comparison with the other sectors of the Southern Ocean (i.e. Indian Ocean sector and Pacific sector), the *ASSO* is colder, more productive (in terms of primary production), and therefore sustaining larger densities of Antarctic krill (*Euphausia superba*) [16]. The abundant availability of Antarctic krill is key to the subsistence of various Antarctic and seasonally visiting predator species, such as crabeater seals (*Lobodon carcinophaga*), Adélie penguins (*Pygoscelis adeliae*) and humpback whales [12].

Recent studies using passive acoustic monitoring (PAM) have discovered that at least parts of the Antarctic blue whale and humpback whale populations even remain in the *ASSO* during austral winter [11,20]. Large-scale trends of the humpback whale distribution, however, remain unexplored. Particularly, in the oceanic regions of the *ASSO*, the distribution patterns of humpback whales are to date largely unknown, although these regions are assumed to be the main migratory destinations of humpback whales from the South Atlantic [21–24].

Technological advances in the fishing industry and predicted climate change might open up new opportunities for the krill fishery shifting fishing grounds further south, where most favourable krill habitats are located [25,26]. Insights into spatio-temporal patterns in the distribution of humpback whales throughout the *ASSO* are therefore of crucial importance for effective management and conservation planning, e.g. by the International Whaling Commission [24]. Furthermore, the scientific community has also proposed the establishment of a Marine Protected Area (MPA) in the Weddell Sea, which aims to also include areas ecologically relevant to large marine predators [25]. Baseline data on the distribution and abundance of species that rely on the resources provided by the Weddell Sea area, such as humpback whales, are crucial for the planning and eventually also the approval of an MPA in the *ASSO*.

This study aims to investigate the year-round distribution of humpback whales over the full spatial range of the *ASSO* by analysing the passive acoustic data collected by a network of 12 simultaneously recording receivers. Humpback whales are a highly vocal species producing sounds on the breeding and feeding grounds as well as during migration, which makes them a suitable species for PAM-based studies [20,27,28]. Through the analysis of a spatially extensive dataset from the *ASSO*, we will explore the spatio-temporal variability in the occupancy of potential feeding areas in the *ASSO* by Southern Hemisphere humpback whales.

# 2. Material and methods

## 2.1. Passive acoustic data

Humpback whale acoustic behaviour was investigated using data from 12 recording positions throughout the *ASSO* (table 1 and figure 1), which recorded simultaneously in 2013 (figure 2). Passive acoustic recordings were obtained using SonoVaults (Develogic GmbH, Hamburg; Reson TC4037-3 hydrophone, -193 dB re1 V μPa$^{-1}$ hydrophone sensitivity, 48 dB amplification gain, 24 bits resolution) operated on a continuous recording scheme and with a sampling rate of 5333 or 9600 Hz (table 1). The recorders were deployed as part of oceanographic moorings with multiple instruments installed on a vertical line which usually extended to 800 m as the shallowest depth (to avoid being damaged by drifting icebergs; except for the mooring position W12 off Elephant Island, where the water depth was only 300 m) (see also [29–31] for more information on the HAFOS moorings).

## 2.2. Automatic detection and classification of humpback whale vocalizations

All available passive acoustic data were processed by the 'low-frequency detection and classification system' (LFDCS) developed by Baumgartner & Mussoline [32] and a custom-made acoustic-context filter to detect humpback whale acoustic presence on an hourly basis. LFDCS was set up with a customized call library based on the most common vocalization types of humpback whales and other acoustically abundant Antarctic marine mammal species (i.e. Antarctic minke whale (*Balaenoptera bonaerensis*), killer whale (*Orcinus orca*), Weddell seal (*Leptonychotes weddellii*), crabeater seal, leopard seal (*Hydrurga leptonyx*) and Ross seal (*Ommatophoca rossii*)) [27,32–37]. Parameter settings and thresholds of LFDCS and the acoustic context filter were tuned employing multiple test datasets to optimize the automatic detection of humpback whale vocalizations to the requirements of this study. Detailed information on set-up and test runs of the automatic detection process are provided in the electronic supplementary material. Resulting detected hours with presumed humpback whale acoustic presence are termed *presumed Humpback Whale Presence* (*pHWP*) hereinafter.

## 2.3. Manual post-processing of detection results

To limit the temporal effort of manual post-processing, only even *pHWP* hours (i.e. hours starting at 00.00, 02.00, 04.00, 06.00, 08.00, 10.00, 12.00, 14.00, 16.00, 18.00, 20.00, 22.00) were included in the further analysis. We evaluated if subsampling only the even hours would not affect the results by performing comparative analyses for two recorders (from 2011) for which all hours were manually analysed. From this full dataset, only even hours were subsampled and the acoustic presence at odd hours was interpolated (condition: two consecutive even hours with acoustic presence determines acoustic presence in intermediate odd hour). When comparing the interpolated results with the original results, similarity between the subsampled-interpolated and full datasets was above 95%. Therefore, acoustic presence in consecutive even hours in the large majority of cases indicates acoustic presence in the intermediate odd hour. Given that the number of acoustic presence hours is underestimated, i.e. approximately halved, our results are all presented as proportions of hours per day or per month. Four human analysts revised even *pHWP* hours visually and aurally for the presence of humpback whale vocalizations by creating spectrograms in Raven Pro 1.5 (Hann Window, 1025–1790 window size, 80% overlap, 2048 DFT size; Bioacoustics Research Program 2014). Spectrograms were screened for humpback whale vocalizations by viewing windows of 60 s duration, spanning 0 to 1.80 kHz. Hours with confirmed humpback whale acoustic presence (herein referred to as *confirmed Humpback Whale Presence*; *cHWP*) could contain both humpback whale social calls and humpback whale song. The level of agreement in manually classifying *cHWP* and false-positive hours between the principal analyst and the other three analysts was calculated on

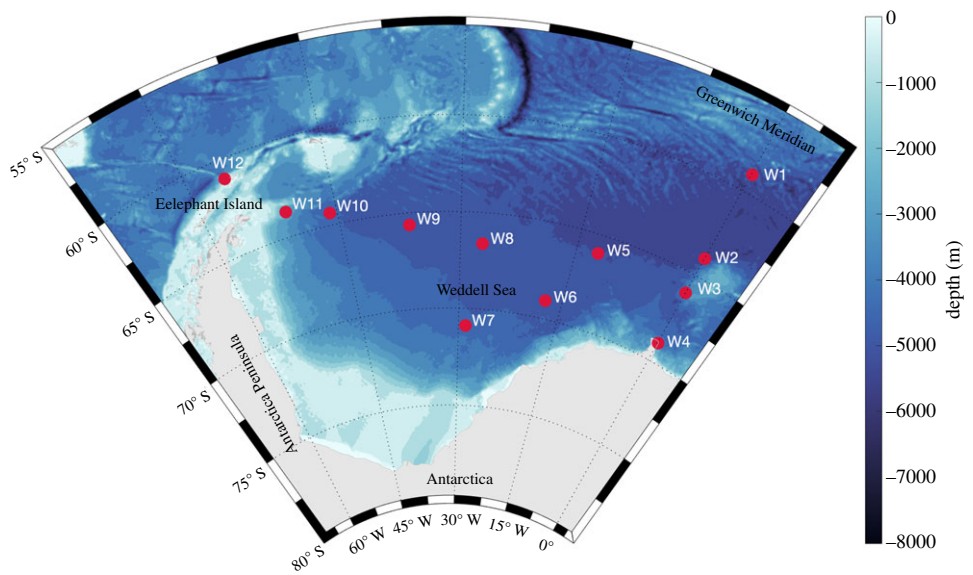

**Figure 1.** Bathymetric map of the *ASSO* and the geographical positions of the 12 bottom-moored recorders included in this study.

**Table 1.** Information on passive acoustic recordings included in the dataset. For reference to earlier publications, the original mooring ID is listed in brackets.

| mooring ID | latitude | longitude | recorder ID | sampling frequency (Hz) | deployment depth (m) |
|---|---|---|---|---|---|
| W1 (AWI227) | 59 2.82° S | 000 5.78° E | SV1025 | 5333 | 1020 |
| W2 (AWI229) | 63 59.85° S | 000 1.84° E | SV1010 | 5333 | 998 |
| W3 (AWI230) | 66 2.01° S | 000 3.12° E | SV1009 | 5333 | 949 |
| W4 (AWI232) | 68 59.94° S | 000 4.38° E | SV1011 | 5333 | 958 |
| W5 (AWI248) | 65 58.09° S | 012 15.12° W | SV1013 | 5333 | 1081 |
| W6 (AWI245) | 69 3.480° S | 017 23.32° W | SV1012 | 5333 | 1065 |
| W7 (AWI249) | 70 53.55° S | 028 53.47° W | SV1014 | 5333 | 1085 |
| W8 (AWI209) | 66 36.45° S | 027 7.26° W | SV1028 | 5333 | 1007 |
| W9 (AWI208) | 65 37.23° S | 036 25.32° W | SV1030 | 5333 | 956 |
| W10 (AWI217) | 64 22.94° S | 045 52.12° W | SV1020 | 5333 | 960 |
| W11 (AWI207) | 63 42.09° S | 050 49.61° W | SV1033 | 9600 | 1012 |
| W12 (AWI251) | 61 1.07° S | 055 58.67° W | SV1008 | 5333 | 212 |

varying test datasets of at least 150 *pHWP* hours (presented in Results). Hourly humpback whale acoustic presences were transformed into proportion of *cHWP* hours per day. Proportions of *cHWP* hours per day were averaged per month and respective standard deviations were calculated or the monthly acoustic presence was calculated as the number of *cHWP* hours per month divided by the total number of recording hours of the respective month.

## 2.4. Sea ice data

The sea ice concentration data used for this study were extracted from: a combination of satellite sensor data from the Nimbus-7 Scanning Multichannel Microwave Radiometer (SMMR), the Defense Meteorological Satellite Program (DMSP) -F8, -F11 and -F13 Special Sensor Microwave/Im rs (SSM/Is) and the DMSP-F17 Special Sensor Microwave Imager/Sounder (SSMIS), with a grid size of 25 km [38]. The data were used to calculate the daily sea ice concentration of the area within 50 km radius around each recording location, with the Daily Antarctic Sea Ice Concentration packages in Matlab

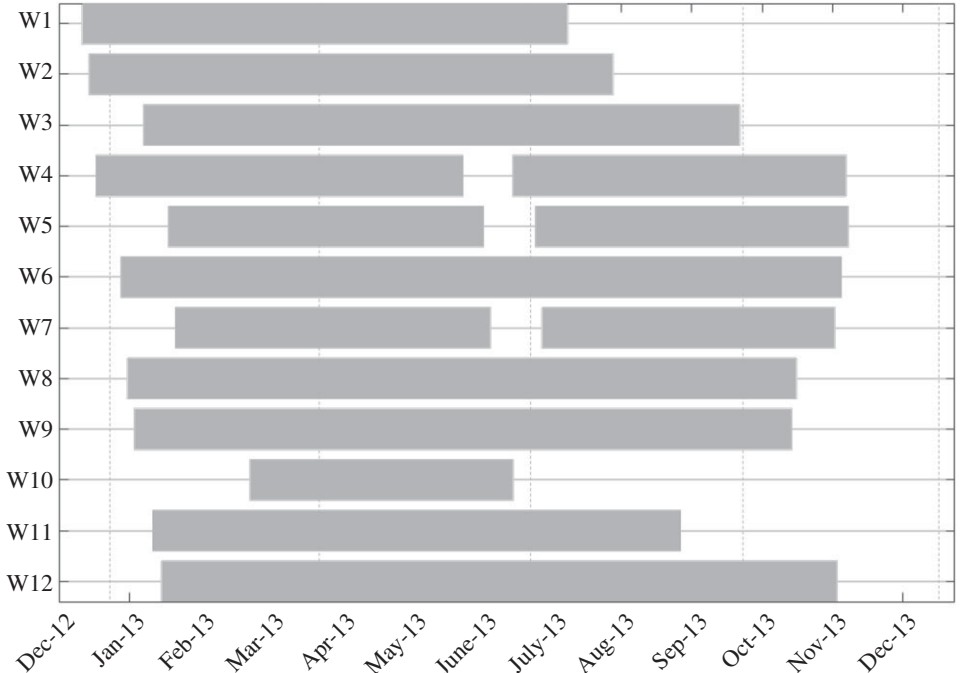

**Figure 2.** Timeline showing the availability of passive acoustic data collected throughout 2013 for the 12 recording positions in the *ASSO*.

[39]. The radius of 50 km was chosen because the acoustic range of humpback whales in the *ASSO* was estimated at 2–78 km [20]. Additionally, the data were used to calculate monthly averages of sea ice concentrations for the *ASSO* and plotted as maps with the Antarctic Mapping Tools and Daily Antarctic Sea Ice Concentration packages in Matlab [39,40]. In order to test for correlations between humpback whale acoustic presence and the local sea ice concentration, the Pearson correlation coefficient was calculated for four different temporal regimes: Comparing monthly averages, comparing three-monthly averages starting in January (i.e. JFM, AMJ, etc.), comparing three-monthly averages starting in February (i.e. FMA, MJJ, etc.), and comparing three-monthly averages starting in March (i.e. MAM, JJA, etc.).

# 3. Results

In total, 74 628 h of recordings were processed, of which 13 049 were *pHWP* hours. Roughly half of these hours were post-processed by human analysts and, summing all recording locations, 983 h were verified as *cHWP* hours (table 2). Among the four analysts, the level of agreement in classifying *cHWP* or false positive hours was between 93% and 97%.

## 3.1. Spatial pattern

During austral summer and autumn (January–June) in 2013, nine of the 12 recording positions recorded humpback whale vocalizations (table 2). At the positions W10, W11 and W7 humpback whale acoustic presence could not be confirmed in 2013 (i.e. considering only the even recording hours were included in the analyses; table 2 and figure 3). At most recording positions (W9, W8, W6 and W4), the monthly acoustic presence of humpback whales was not higher than 10% (figure 3). The recording positions W5 and W12 registered monthly humpback whale acoustic presences of up to 20% and at the recording position off Elephant Island (W12), humpback whales were acoustically active during all recorded months of the year 2013 (figure 3). The highest monthly acoustic presences of humpback whales (i.e. greater than 20%) were confirmed at the four recording positions W3, W2 and W1 on the Greenwich Meridian (figure 3). Monthly acoustic presences of more than 10% were only registered in areas without sea ice cover, and, in most cases, monthly acoustic presences of 0% were only registered in areas with a sea ice concentration of at least 50%. In the central Weddell Sea, humpback whales were only sporadically acoustically present (i.e. less than or equal to 10%), which presumably was

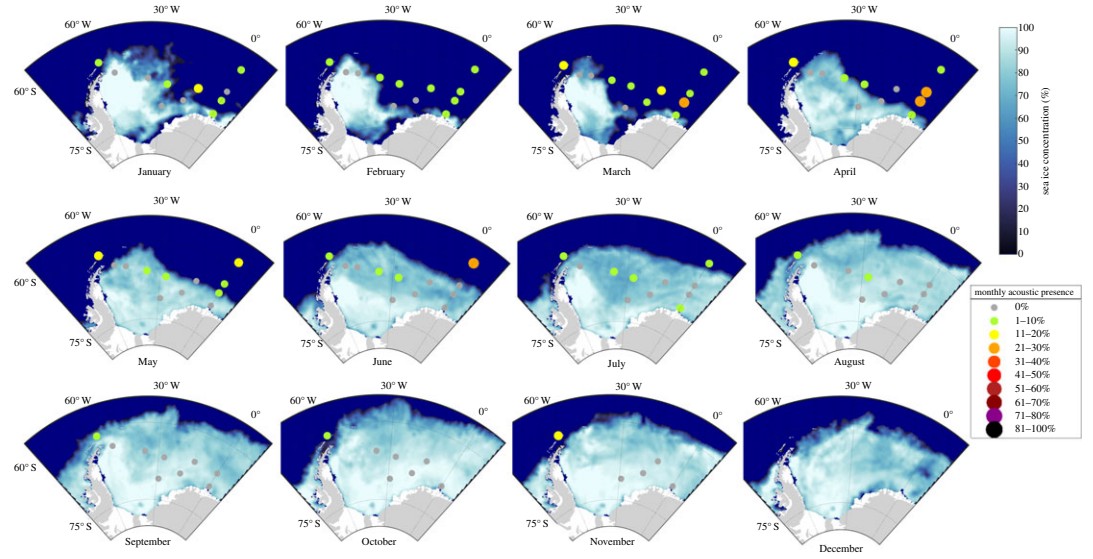

**Figure 3.** Percentage of acoustic presence of humpback whales in the *ASSO* averaged per recording location and month for the year 2013. Size and colour of dots indicates a respective range of percentage of hours per month with humpback whale acoustic presence. The monthly averaged sea ice concentrations are depicted at a $25 \times 25$ km resolution.

**Table 2.** Overview of recording hours, presumed humpback whale acoustic presence (*pHWP*) hours, post-processed hours and confirmed humpback whale acoustic presence (*cHWP*) hours per recording location and as an overall sum.

| mooring ID | total hours | *pHWP* hours | hours post-processed | *cHWP* hours |
|---|---|---|---|---|
| W1 | 5140 | 584 | 284 | 200 |
| W2 | 5538 | 628 | 306 | 157 |
| W3 | 6316 | 826 | 428 | 203 |
| W4 | 7386 | 1958 | 993 | 23 |
| W5 | 6649 | 1440 | 739 | 77 |
| W6 | 7630 | 163 | 85 | 10 |
| W7 | 6424 | 312 | 157 | 0 |
| W8 | 7077 | 1639 | 823 | 16 |
| W9 | 6973 | 1159 | 594 | 46 |
| W10 | 2767 | 452 | 230 | 0 |
| W11 | 5558 | 923 | 460 | 0 |
| W12 | 7170 | 2965 | 1476 | 251 |
| total | 74 628 | 13 049 | 6575 | 983 |

related to the fact that the area was covered with sea ice over extended periods (figure 3). At positions W11, W10 and W7, which were covered by sea ice almost year-round, humpback whales were acoustically absent throughout 2013.

## 3.2. Intra-annual temporal pattern

From January until May 2013, at least 50% of recording positions registered humpback whale acoustic presences (figures 3 and 4). In March 2013, humpback whales were acoustically present at the largest proportion of the recording positions (9 out of 12). At recording position W6, for example, humpback whales were acoustically present exclusively during a continuous period of 4 days, between 18 March 2013 and 21 March 2013. At the recording position W5, humpback whales were acoustically present in January, February and March 2013 (figure 4). April 2013 was the month with the highest

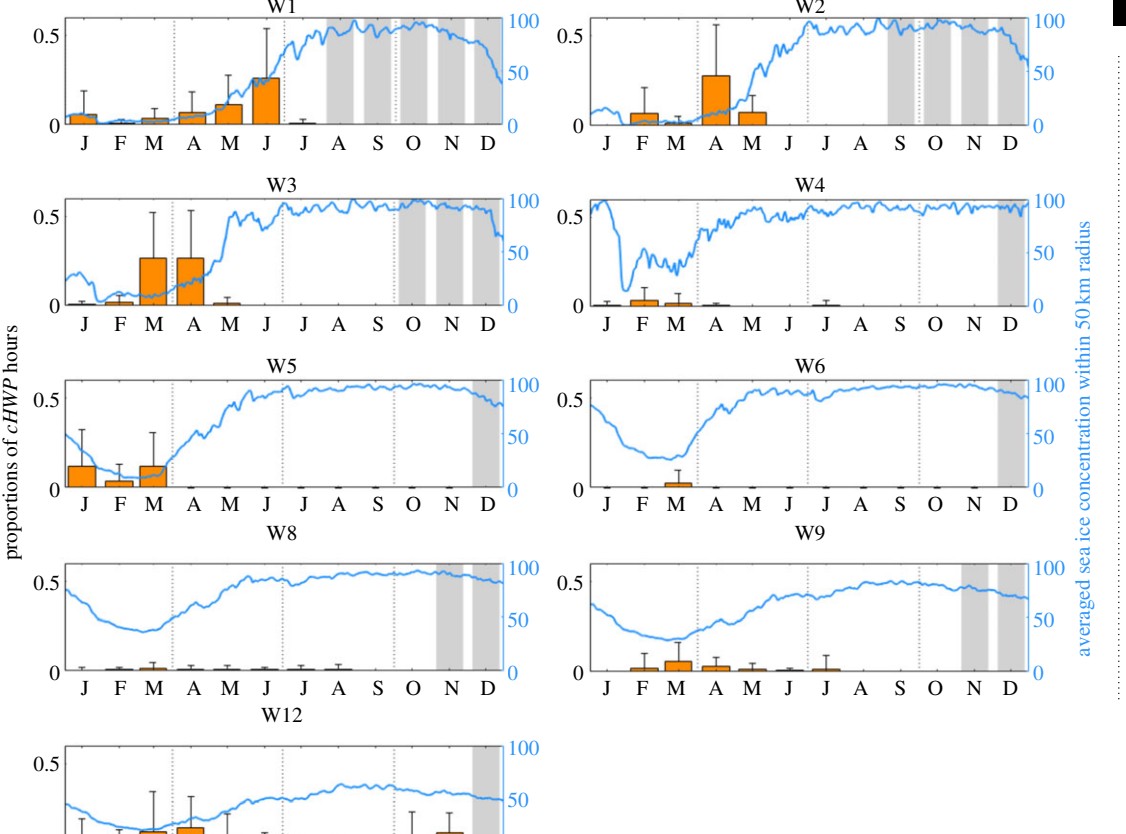

**Figure 4.** Average proportion of confirmed humpback whale presence (*cHWP*) hours per month at the nine recording positions, W1, W2, W3, W4, W5, W6, W8, W9 and W12 from January until December 2013 (orange bars). Vertical error bars show the respective standard deviations and continuous grey bars represent months without recording data. The blue solid lines and the right *y*-axis depict the daily averaged sea ice concentration per location within a 50 km radius (average sea ice concentration data for recording position W12 has to be interpreted with caution, as data pixels along the coast, covering both oceanic and ice shelf areas, remain masked in the dataset used, i.e. are biased towards high values). At three recording positions (W7, W10 and W11) humpback whales were acoustically absent, graphs are therefore not displayed.

proportion of *cHWP* hours summed over all recording positions. Off Elephant Island (W12) the peak periods for humpback whale acoustic presence were March until May and October and November (figure 4). During the months January and February and June until September only sporadic acoustic presence (i.e. only single hours) was confirmed at the recording position W12 (figure 4). Similarly, sporadic acoustic presence of humpback whales was registered for January/February until August/July at recording positions W8 and W9, respectively (figure 3). At the recording positions W1 to W3 at the Greenwich Meridian the acoustic presence of humpback whales was strongly seasonal: humpback whales were acoustically present between January and July with peak periods in March until June (depending on the position; figure 4). By contrast, at the southernmost recording position at the Greenwich Meridian (W4), *cHWP* hours were confirmed sporadically in the months January, February, March, April and July (figure 4).

## 3.3. Diurnal pattern

The data of most recording positions did not show diurnal patterns in humpback whale acoustic presence when comparing the proportions of (even) hours of the day with confirmed humpback whale acoustic presence per month against each other. For example, at the recording position W12 off Elephant Island (figure 5), but also at the positions W3, W4, W5, W6, W8 and W9, humpback whales were acoustically present during seemingly random hours of the day (W7, W10 and W11 did not record humpback whale vocalizations at all). Only at the recording positions W1 and W2 a weak diurnal pattern can be detected during the months May and June (figure 5). During these months,

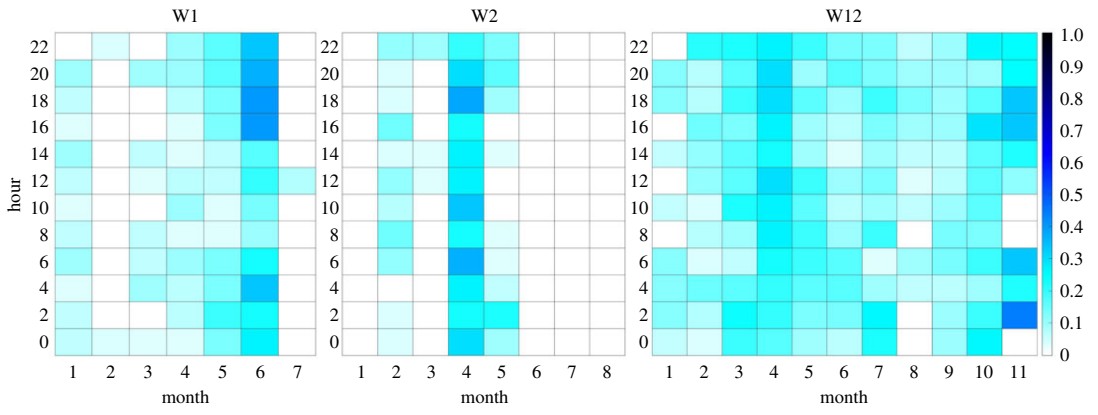

**Figure 5.** Diurnal pattern per month at the recording positions W1, W2 and W12. At the three recording positions W7, W10 and W11 humpback whales were acoustically absent, graphs are therefore not displayed. The diurnal pattern at W3, W4, W5, W6, W8 and W9 was similarly random as at W12, therefore these graphs are also not displayed. Proportions of confirmed humpback whale presence (*cHWP*) hours were calculated for each hour of the day and month of the year 2013. The *y*-axes only display even hours, because only even hours were analysed in this study.

humpback whales were less acoustically active in the morning and during midday (i.e. from 06.00/08.00 until 14.00/16.00; figure 5).

## 3.4. Spatio-temporal trends in relation to sea ice

The spatial pattern of humpback whale acoustic presence in the *ASSO* can be reduced to a longitudinal and a latitudinal trend. The longitudinal trend was characterized by minimal average proportions of *cHWP* hours at the central longitudes of the study area, while at the western and eastern edges of the study area the highest average proportions of *cHWP* hours were recorded. In turn, the latitudinal trend was clearly linear, with increasing average proportions of hourly acoustic presences at decreasing latitudes (i.e. from south to north). Both spatial trends are connected to the spatial extent of the sea ice cover which temporally opened up especially at the western, eastern and northern edges of the *ASSO*, but which was present year-round in the southern-central part of the Weddell Sea (figure 3).

The intra-annual temporal pattern of humpback whale acoustic presence in the *ASSO* was not clearly driven by sea ice concentration. Monthly and three-monthly averages of humpback whale acoustic presence were only weakly correlated with the local sea ice concentration (within a 50 km radius). The pronounced seasonal acoustic presence of humpback whales at the Greenwich Meridian (three oceanic recording positions W1–W3) nevertheless seems to be connected to the presence of sea ice. During the rapid decrease in sea ice concentration in the beginning of summer, humpback whale acoustic presence was generally low at the Greenwich Meridian (figure 4). The first acoustic activity of humpback whales in the season was within 1 day and 56 days after the sea ice concentration dropped below 15% (for definition sea ice edge, see [41]). At all three oceanic recording positions (i.e. W1–W3), the proportion of *cHWP* hours peaked simultaneously with the rapid increase of the sea ice concentration in late summer/autumn (figure 4). The last acoustic activity of humpback whales in the season was within 39 to 67 days after the sea ice concentration exceeded 15%. At all recording locations on the Greenwich Meridian, the proportion of *cHWP* hours declined when the sea ice concentration exceeded 50% (figure 4).

# 4. Discussion

## 4.1. Spatial distribution

Our results confirm earlier observations that the *ASSO* is likely to form an important feeding area for humpback whales from the South Atlantic (breeding stock A off the coast of Brazil and stock B off the coast of Angola/Gabon, see [14]). Humpback whales are known to migrate between ocean basins and migration from the eastern South Pacific and western Indian Ocean into the *ASSO* has been suggested as well [6,14,42]. The highest proportions of *cHWP* hours were recorded at the eastern and western edges of our study area, which are the direct longitudinal extensions of the South American and

African continents. In the Southern Hemisphere, migrating humpback whales are often observed to travel along or close to coastlines, where coastal fronts are thought to aid navigation and provide chances for opportunistic feeding [23,43–45]. The eastern and western acoustic hotspots in our data could therefore reflect humpback whale migratory routes along the eastern/western coastline of South America/South Africa extending south towards the Antarctic continent. Satellite tracking studies targeting humpback whales off Brazil, Gabon and South Africa revealed possible summer feeding destinations north of 60° S in the waters around South Georgia, the South Sandwich Islands and Bouvet Island, but did not register any movements inside the Southern Ocean [21–24]. Besides the favourable position in terms of distance to breeding areas, the eastern and western edges of the *ASSO* could also present areas of elevated food availability; the coastal areas around the northern part of the Western Antarctic Peninsula are known for high densities of Antarctic krill and smaller krill hotspots can also be found along the Greenwich Meridian [19,46].

Alternatively, the observed longitudinal trend could reflect an underlying latitudinal trend. At the eastern and western edges of the *ASSO,* data collection was biased towards lower latitudes, where generally more calls were recorded compared with the higher latitude recording sites. Our data show a clear latitudinal trend with the highest proportions of *cHWP* hours at the most northern recording positions. There are several possible explanations if this trend is real. First, it could be related to the trade-off between the cost of migration and the energetic gain of feeding in high-latitude waters [47,48]. Southern Hemisphere humpback whales migrate southward with the retreating sea ice edge to search for high densities of near-surface swarms of euphausiids in order to maximize their energy intake [47,48]. To minimize the energetic effort, they possibly only travel as far south as necessary to restore energy reserves. An alternative explanation for the observed latitudinal trend is that humpback whales decrease their vocal activity as they move south, e.g. determined by decreasing hormone levels in spring [49]. Humpback whales are sometimes sighted south of 70° S, indicating that single whales are roaming these waters, but might not be acoustically active during this time [50]. Further collection of passive acoustic data over a longer period of time (i.e. longer than one year) combined with visual data are underway and will make it possible to draw further conclusions on these observations.

## 4.2. Seasonal and diurnal patterns in humpback whale acoustic presence

Humpback whale movement strategies in the *ASSO* are probably optimized in terms of the energy gain and costs, most likely driving intra-annual and potentially even diurnal patterns of acoustic presence in the *ASSO*. Individual humpback whales are likely to adapt their habitat selection and migratory behaviour on the feeding grounds based on their life stage, reproductive status and body condition, as has been confirmed for many baleen whale species [7,51]. This diverse repertoire of migratory behaviour and the ability to adapt to the local environment probably explains the observed seasonal fluctuation in humpback whale acoustic absence and presence throughout the study area.

Summarizing all recording positions, our data indicate humpback whale presence in the *ASSO* during all months of the year, except December. However, for all locations, overall data coverage for December was poor (only a few days during December 2012) which could have affected detection probability of calls. In January and February, also only low proportions of *cHWP* hours were recorded at all recording locations, while overall data coverage was good for these months. These months could either be the time with the fewest or no humpback whales present in the *ASSO*, or represent a period during which whales do not or only rarely vocalize. From ship-based sighting surveys, it is known that humpback whales are regularly sighted in the *ASSO* from December to February [50,52–60], indicating that humpback whales are physically present in the area but may be less vocal during this time. This finding temporally matches the singing pause registered for Northern Hemisphere humpback whales from June to August, when humpback whales probably concentrate on feeding activities to rapidly restore their energy budgets [61].

The virtually basin-wide and near year-round acoustic presence of humpback whales reported in this study suggests that individuals frequenting this area may regularly deviate from the traditional migration model. During austral winter, at least some humpback whales seem to remain in areas of the *ASSO* without sea ice cover, e.g. the waters around Elephant Island or coastal polynyas close to the Antarctic continent (recording position W4 and also see [20]). Similar to what has been reported for humpback whales from other ocean basins, humpback whales migrating in and out of the *ASSO* are likely to exhibit diverse migration strategies [20], potentially including sex- and age-dependent differences in timing of migration, as well as the complete omission of migration during some years [7,9,51,62,63]. During March, humpback whales were acoustically present at the most recording locations

simultaneously (nine out of 12). March could be the time of the year, when most humpback whales, including all sex and age classes, are arriving at the feeding areas in the Southern Ocean, which in turn causes a higher spatial dispersal of feeding individuals or groups to avoid competition. April, May, June were the months with the highest proportions of *cHWP* hours recorded during this study. A high proportion of the acoustic activity during these months was attributed to singing humpback whale males (preliminary analyses show up to 50% of the vocal activity consists of song which is known to be exclusively produced by males; Schall *et al.* unpublished data). The austral fall forms part of the 'pre-breeding shoulder season', which is the period preceding the breeding season. During this time humpback whale males start singing before or while migrating to the breeding grounds, presumably to improve their chances of mating success [5]. During the months August/September generally low proportions of *cHWP* hours were recorded at all recording locations. August and September most likely represent the months during which fewest individual humpback whales are present in the *ASSO*, because most (vocally active) individuals spend this time at their low-latitude breeding grounds [10,64,65].

Our recordings did not exhibit any clear diurnal pattern for the acoustic activity of humpback whales in the *ASSO*, which suggests opportunistic sound production during random times of the day. During austral summer, humpback whales might prioritize restoring their energy reserves in a time-efficient manner and might be concentrating most of their activities on feeding and searching for prey. In the waters off the western Antarctic Peninsula, tagging studies have shown that humpback whales follow a diel feeding pattern, with most feeding dives occuring at night when krill swarms are closer to the surface [66]. This vertical migration in Antarctic krill has been described for various regions in the Southern Ocean, although the pattern is not consistent for all regions across the Southern Ocean (see [19] for overview). Humpback whales in the *ASSO* might therefore feed and vocalize rather opportunistically, adapting their behaviour to changes in local prey availability and the presence of conspecifics.

## 4.3. Spatio-temporal trends and sea ice

The estimated correlation between humpback whale acoustic presence and sea ice concentration was weak. It cannot be excluded that this weak correlation is a consequence of inaccuracies in the sea ice concentration data, which might be biased towards high values due to merging ice shelf areas with oceanic areas for pixels intersecting the coast [38]. The recordings from the Greenwich Meridian suggest that humpback whales moved south, following the retreating sea ice edge. Humpback whales generally seem to prefer open water or larger ice-free areas within the sea ice (i.e. polynyas), which is probably related to the easier access to ice-free space for breathing [20,67]. Along the sea ice edge, humpback whale feeding groups could also be exploiting the high densities of krill, characteristic for the marginal ice zone [19,68]. The dynamic interactions between nutrient supply by melting sea ice, open water fuelling primary production and sea ice as a key habitat for juvenile krill [69–71] influence prey availability for humpback whales in the *ASSO* in a complex spatio-temporal arrangement [72].

## 4.4. The *ASSO* humpback whale feeding ground

The *ASSO* is probably a feeding ground for at least two humpback whale breeding populations [6,14,42]. The distinct peaks of acoustic activity detected at the eastern and western edges and potentially even the differentiation of temporal patterns (i.e. the western edge with a rather continuous acoustic presence pattern and the eastern edge with a seasonal acoustic presence pattern) may be reflective of the presence of two distinct humpback whale populations. The spatial segmentation of the *ASSO* feeding ground for the distinct humpback whale breeding populations as well as the potential overlap in the occupied area among these populations represents baseline knowledge necessary for efficient stock management and deserves further investigation. Our study, among many others (e.g. [11,20,73]), has proven remote PAM as very effective for the study of highly mobile marine mammal species in the Southern Ocean. The more detailed analysis of humpback whale acoustic recordings can provide further information on male singing behaviour, which is thought to a be a population-specific reproductive display [74,75].

The attribution of specific feeding grounds to the humpback whale populations in the Southern Hemisphere, as well as the level of connectivity among these distinct breeding stocks are still largely unresolved [24]. Both the political difficulties of implementing dynamic conservation strategies for migratory species as well as the need to estimate the ecological capacity of the *ASSO* food web for krill fishery stock management, would profit from insights in the distribution range for individual humpback whale stocks. Ongoing investigations of humpback whale songs in the *ASSO* are therefore aimed at obtaining such fundamental insights into the population-specific distribution patterns within this important Southern Ocean feeding ground.

Ethics. Permission for mooring installation was granted to the Alfred Wegener Institute, Helmholtz-Zentrum for Polar- und Meeresforschung by the Federal Environment Office (Umweltbundesamt UBA): Expedition ANT - XXIX/2 UBA permit no. I 3.5-94003-3/286, Expedition PS89 UBA permit no. II 2.8-94003-3/324.

Data accessibility. Data on the acoustic presence of humpback whales in the *ASSO* is uploaded on Dryad (https://doi.org/10.5061/dryad.ncjsxkss0 [76]). Sea ice concentration data can be downloaded from MathWorks and the National Snow and Ice Data Centre (https://www.mathworks.com/matlabcentral/fileexchange/50126-daily-antarctic-sea-ice-concentration) (https://nsidc.org/data/NSIDC-0051/versions/1). Visual sighting data from the *ASSO* are uploaded on PANGAEA (https://doi.org/10.1594/PANGAEA.896842; https://doi.org/10.1594/PANGAEA.840382; https://doi.org/10.1594/PANGAEA.819861; https://doi.org/10.1594/PANGAEA.819862; https://doi.org/10.1594/PANGAEA.819863; https://doi.org/10.1594/PANGAEA.819866; https://doi.org/10.1594/PANGAEA.783806; https://doi.org/10.1594/PANGAEA.760340).

Authors' contributions. E.S. analysed the data and wrote the manuscript. K.T. participated in some data collection and helped draft the manuscript. O.B. participated in collecting data and coordinated the study. G.G. guided the analysis and helped draft the manuscript. S.S. collected all the data. I.V.O. coordinated the study, collected part of the data and helped draft the manuscript. All the authors reviewed and contributed to the final document edits. All the authors gave the final approval for publication.

Competing interests. We declare we have no competing interests.

Funding. Funding sources are equivalent to affiliations.

Acknowledgements. Thanks to Develogic GmbH, Hamburg, to the logistics department of the Alfred Wegener Institute, Bremerhaven, the mooring team of the AWI's physical oceanography department, to Reederei F. Laeisz GmbH, Rostock and the crew of RV Polarstern expedition ANT-XXIX/2 and PS89, for their contribution to the development, set-up or maintenance of the passive acoustic recording array. We thank Maria Mallet and Katharina Hiemer for assistance with the manual post-processing of acoustic data and the whole team of the Ocean Acoustics Laboratory for the productive discussions on this study. We also want to thank Mark Baumgartner and Genevieve Davis for the assistance in setting up LFDCS.

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
