## [Reviewer comments · Royal Society Open Science]

Review History

RSOS-201347.R0 (Original submission)

Review form: Reviewer 1

Is the manuscript scientifically sound in its present form?

Yes

Are the interpretations and conclusions justified by the results?

Yes

Is the language acceptable?

Yes

Do you have any ethical concerns with this paper?

No

Have you any concerns about statistical analyses in this paper?

No

Recommendation?

Accept with minor revision (please list in comments)

Comments to the Author(s)

1. Original Submission

1.1 Recommendation

Minor Revision

2. Comments to Author:

Manuscript Number: RSOS-201347

Title: Large-scale spatial variabilities in the humpback whale acoustic presence in the Atlantic sector of the Southern Ocean

Elena Schall, Karolin Thomisch, Olaf Boebel, Gabriele Gerlach, Stefanie Spiesecke, Ilse Van Opzeeland

Overview and general recommendation:

The authors present an interesting study on humpback whales' spatio-temporal activity in the Atlantic Sector of the Southern Ocean based on Passive Acoustic Monitoring data. Understanding humpback whales' habits is relevant to the stock and general ecosystem management, which justifies the effort to improve the knowledge about them. The results bring new information that may support insights about the dynamics of humpback whales populations. Thus, considering the importance of the theme and the approach used, the manuscript fits the journal scope very well.

Nevertheless, after reading the manuscript, I have some remarks and questions that should be addressed and clarified:

Comments:

1. Line 18

"During all months, except December, humpback whale acoustic activity was registered in the ASSO."

The readers should be informed that December was barely sampled. Otherwise, they may misunderstand that there was no humpback whale activity in December. Why was December poorly sampled? How was the logistics to deploy all the moorings? How many cruises? What caused the sampling failures? Logistic problems, electronic malfunctioning, damage due to the ice cover?

2. Lines 89-90, and Table 1 (page 17)

Why was the W11 sampling rate set up to 9,600 Hz while all 11 stations were set up to 5,333 Hz? The authors should provide more information about the hydrophones and setup (sensitivity, digital sample resolution, gain, pre-amp, etc.). Why was the W12 deployed at 212 meters depth, while the others were near 1000 meters? The authors should also provide more information about the mooring line.

3. Lines 105-107

What is the impact of analyzing only even hours? Is the number of cHWP underestimated? What does it mean? Do cHWP in consecutive even hours may indicate cHWP in intermediate odd hours?

4. Lines 130-133: "In order to test for correlations between humpback whale acoustic presence and the local sea ice concentration, the Pearson correlation coefficient was calculated for 4 different temporal regimes: Monthly and three-monthly starting in January, February, or March."

I suggest that the authors rephrase the sentence in order to clarify this topic.

5. Lines 141-142: "During austral summer and autumn in 2013, nine of the 12 recording positions recorded humpback whale vocalizations (Table 2)."

It should be noted that the hydrophone W10 was deployed after summer, and the majority of the hydrophones did not fully sample the summer season. I suggest annotating the months in parentheses after "summer and autumn.". In Figure 1, I also suggest drawing vertical lines to separate the seasons.

6. Lines 154-155: "At positions W11, W10, and W7, which were covered by sea ice almost year-round, humpback whales were acoustically absent throughout 2013."

The information is related to the lines 141-142. They could be merged.

7. Lines 169-172: "At the recording positions W1 through W3 at the Greenwich Meridian the acoustic presence of humpback whales was strongly seasonal: humpback whales were acoustically present between January and July with peak periods in March until June (depending on the position; Figure 4)."

I am not sure if it is wise to affirm that considering the lack of sampling—the W1 station stops in July, W2 in August, and W3 in late September.

8. Line 215: "continents."

9. Line 227: "observed."

10. Line 238: "70°S."

11. Lines 250-254: "Summarizing all recording positions, our data indicate humpback whale presence in the ASSO during all months of the year, except December. December could either be the month with the fewest or no humpback whales present in the ASSO, or represent a period during which whales do not or only rarely vocalize. However, for all locations, overall data coverage for December was poor (only a few days during December 2012) which could have affected detection probability of calls."

Considering data coverage for December was poor, as noted by the authors, the phrase "December could either be the month with the fewest or no humpback whales present in the ASSO, or represent a period during which whales do not or only rarely vocalize." turns into speculation. It is not possible to make conclusions based on a lack of observation due to poor sampling.

12. Line 257: "humpback whales are regularly sighted in the ASSO..."

I suggest adding the references:

DALLA ROSA, L. ; SECCHI, E. R. ; KINAS, P. G. ; SANTOS, M. C. O. ; MARTINS, M. B. ; ZERBINI, A. N. ; BETHLEM, C. . Photo-identification of humpback whales, *Megaptera novaeangliae*, off the Antarctic Peninsula: 1997/98 to 1999/2000. *Memoirs of the Queensland Museum*, v. 47, n.2, p. 555-561, 2001.

SECCHI, E. R.; DALLAROSA, L. ; KINAS, P. G. ; SANTOS, M. C. O. ; ZERBINI, A. N. ; BASSOI, M. ; MORENO, I. B. . Encounter rates of whales around the Antarctic Peninsula with special reference to humpback whales, *Megaptera novaeangliae*, in the Gerlache Strait: 1997/98 to 1999/2000. *Memoirs of the Queensland Museum*, v. 47, n.2, p. 571-578, 2001.

13. Line 265: "the waters around the Elephant Island..."

Please, consider to include a legend on the map (Figure 2).

14. Lines 274-276: "A high proportion of the acoustic activity during these months was attributed to singing humpback whale males (Schall et al. unpublished data).

"attributed", instead of "atributed."

Please, provide more information about the reference Schall et al. since it is used to support an important statement.

15. Figure 4

Please, include panels for W7, W10, and W11.

16. Figure and table captions. Figure 4.

"Average proportion of confirmed humpback whale presence (cHWP)..."

Review form: Reviewer 2

Is the manuscript scientifically sound in its present form?

Yes

Are the interpretations and conclusions justified by the results?

Yes

Is the language acceptable?

Yes

Do you have any ethical concerns with this paper?

No

Have you any concerns about statistical analyses in this paper?

No

Recommendation?

Accept with minor revision (please list in comments)

Comments to the Author(s)

Line 13 - "allows to study" consider changing to "allows the study of"

Line 35 - "extent" to "extend"

Lines 51-52 - an isolating current - see comment on pdf (Appendix A).

Lines 64-66 - Revise for clarity

Line 88 - Consider revising figure names

Lines 132-133- Revise for clarity

Lines 147-148 - Revise for clarity - Including pHWP? Because according to cHWP, no humpback whales were recorded at any recorders.

Lines 166-169 - Revise for clarity

Lines 177-179 - I would specify the reason that W3 - W11 were not included in Figure 4. At this point it is only intuitive after reading the next sentence. Additionally you say that W2 show random HW diurnal presence yet still include it in Figure 4 which may be considered slightly confusing with regards to which recorders were included and which not.

Lines 270-271 - Revise for clarity

Line 365 - Check reference format inconsistencies

Lines

Decision letter (RSOS-201347.R0)

Dear Ms Schall

On behalf of the Editors, we are pleased to inform you that your Manuscript RSOS-201347 "Large-scale spatial variabilities in the humpback whale acoustic presence in the Atlantic sector of the Southern Ocean" has been accepted for publication in Royal Society Open Science subject to minor revision in accordance with the referees' reports. Please find the referees' comments along with any feedback from the Editors below my signature.

Please submit your revised manuscript and required files (see below) no later than 7 days from today's (ie 27-Oct-2020) date. Note: the ScholarOne system will 'lock' if submission of the revision is attempted 7 or more days after the deadline. If you do not think you will be able to meet this deadline please contact the editorial office immediately.

on behalf of Prof Pete Smith (Subject Editor)
openscience@royalsociety.org

Associate Editor Comments to Author:

The reviewers have recommended that your paper be accepted following a number of minor revisions. Please ensure that you fully respond to these.

Reviewer comments to Author:
Reviewer: 1

Comments to the Author(s)
1. Original Submission

1.1 Recommendation

Minor Revision

2. Comments to Author:

Manuscript Number: RSOS-201347

Title: Large-scale spatial variabilities in the humpback whale acoustic presence in the Atlantic sector of the Southern Ocean

Elena Schall, Karolin Thomisch, Olaf Boebel, Gabriele Gerlach, Stefanie Spiesecke, Ilse Van Opzeeland

Overview and general recommendation:

The authors present an interesting study on humpback whales' spatio-temporal activity in the Atlantic Sector of the Southern Ocean based on Passive Acoustic Monitoring data. Understanding humpback whales' habits is relevant to the stock and general ecosystem management, which justifies the effort to improve the knowledge about them. The results bring new information that may support insights about the dynamics of humpback whales populations. Thus, considering the importance of the theme and the approach used, the manuscript fits the journal scope very well.

Nevertheless, after reading the manuscript, I have some remarks and questions that should be addressed and clarified:

Comments:

1. Line 18

"During all months, except December, humpback whale acoustic activity was registered in the ASSO."

The readers should be informed that December was barely sampled. Otherwise, they may misunderstand that there was no humpback whale activity in December. Why was December poorly sampled? How was the logistics to deploy all the moorings? How many cruises? What caused the sampling failures? Logistic problems, electronic malfunctioning, damage due to the ice cover?

2. Lines 89-90, and Table 1 (page 17)

Why was the W11 sampling rate set up to 9,600 Hz while all 11 stations were set up to 5,333 Hz? The authors should provide more information about the hydrophones and setup (sensitivity, digital sample resolution, gain, pre-amp, etc.). Why was the W12 deployed at 212 meters depth, while the others were near 1000 meters? The authors should also provide more information about the mooring line.

3. Lines 105-107

What is the impact of analyzing only even hours? Is the number of cHWP underestimated? What does it mean? Do cHWP in consecutive even hours may indicate cHWP in intermediate odd hours?

4. Lines 130-133: "In order to test for correlations between humpback whale acoustic presence and the local sea ice concentration, the Pearson correlation coefficient was calculated for 4 different temporal regimes: Monthly and three-monthly starting in January, February, or March."

I suggest that the authors rephrase the sentence in order to clarify this topic.

5. Lines 141-142: "During austral summer and autumn in 2013, nine of the 12 recording positions recorded humpback whale vocalizations (Table 2)."

It should be noted that the hydrophone W10 was deployed after summer, and the majority of the hydrophones did not fully sample the summer season. I suggest annotating the months in

parentheses after "summer and autumn.". In Figure 1, I also suggest drawing vertical lines to separate the seasons.

6. Lines 154-155: "At positions W11, W10, and W7, which were covered by sea ice almost year-round, humpback whales were acoustically absent throughout 2013."

The information is related to the lines 141-142. They could be merged.

7. Lines 169-172: "At the recording positions W1 through W3 at the Greenwich Meridian the acoustic presence of humpback whales was strongly seasonal: humpback whales were acoustically present between January and July with peak periods in March until June (depending on the position; Figure 4)."

I am not sure if it is wise to affirm that considering the lack of sampling – the W1 station stops in July, W2 in August, and W3 in late September.

8. Line 215: "continents."

9. Line 227: "observed."

10. Line 238: "70°S."

11. Lines 250-254: "Summarizing all recording positions, our data indicate humpback whale presence in the ASSO during all months of the year, except December. December could either be the month with the fewest or no humpback whales present in the ASSO, or represent a period during which whales do not or only rarely vocalize. However, for all locations, overall data coverage for December was poor (only a few days during December 2012) which could have affected detection probability of calls."

Considering data coverage for December was poor, as noted by the authors, the phrase "December could either be the month with the fewest or no humpback whales present in the ASSO, or represent a period during which whales do not or only rarely vocalize." turns into speculation. It is not possible to make conclusions based on a lack of observation due to poor sampling.

12. Line 257: "humpback whales are regularly sighted in the ASSO..."

I suggest adding the references:

DALLA ROSA, L. ; SECCHI, E. R. ; KINAS, P. G. ; SANTOS, M. C. O. ; MARTINS, M. B. ;

ZERBINI, A. N. ; BETHLEM, C. . Photo-identification of humpback whales, *Megaptera novaeangliae*, off the Antarctic Peninsula: 1997/98 to 1999/2000. *Memoirs of the Queensland Museum*, v. 47, n.2, p. 555-561, 2001.

SECCHI, E. R.; DALLAROSA, L. ; KINAS, P. G. ; SANTOS, M. C. O. ; ZERBINI, A. N. ; BASSOI, M. ; MORENO, I. B. . Encounter rates of whales around the Antarctic Peninsula with special reference to humpback whales, *Megaptera novaeangliae*, in the Gerlache Strait: 1997/98 to 1999/2000. *Memoirs of the Queensland Museum*, v. 47, n.2, p. 571-578, 2001.

13. Line 265: "the waters around the Elephant Island..."

Please, consider to include a legend on the map (Figure 2).

14. Lines 274-276: "A high proportion of the acoustic activity during these months was attributed to singing humpback whale males (Schall et al. unpublished data).

"attributed", instead of "atributed."

Please, provide more information about the reference Schall et al. since it is used to support an important statement.

15. Figure 4

Please, include panels for W7, W10, and W11.

16. Figure and table captions. Figure 4.

"Average proportion of confirmed humpback whale presence (cHWP)..."

Reviewer: 2

Comments to the Author(s)

Line 13 - "allows to study" consider changing to "allows the study of"

Line 35 - "extent" to "extend"

Lines 51-52 - an isolating current - see comment on pdf

Lines 64-66 - Revise for clarity

Line 88 - Consider revising figure names

Lines 132-133- Revise for clarity

Lines 147-148 - Revise for clarity - Including pHWP? Because according to cHWP, no humpback whales were recorded at any recorders.

Lines 166-169 - Revise for clarity

Lines 177-179 - I would specify the reason that W3 - W11 were not included in Figure 4. At this point it is only intuitive after reading the next sentence. Additionally you say that W2 show random HW diurnal presence yet still include it in Figure 4 which may be considered slightly confusing with regards to which recorders were included and which not.

Lines 270-271 - Revise for clarity

Line 365 - Check reference format inconsistencies

Lines

===PREPARING YOUR MANUSCRIPT===

===PREPARING YOUR REVISION IN SCHOLARONE===

Author's Response to Decision Letter for (RSOS-201347.R0)

See Appendix B.

Decision letter (RSOS-201347.R1)

Dear Ms Schall,

It is a pleasure to accept your manuscript entitled "Large-scale spatial variabilities in the humpback whale acoustic presence in the Atlantic sector of the Southern Ocean" in its current form for publication in Royal Society Open Science. The comments of the reviewer(s) who reviewed your manuscript are included at the foot of this letter.

Follow Royal Society Publishing on Twitter: @RSocPublishing
Follow Royal Society Publishing on Facebook:
<https://www.facebook.com/RoyalSocietyPublishing.FanPage/>

Read Royal Society Publishing's blog:
<https://royalsociety.org/blog/blogsearchpage/?category=Publishing>

Appendix A**ROYAL SOCIETY
OPEN SCIENCE****Large-scale spatial variabilities in the humpback whale
acoustic presence in the Atlantic sector of the Southern
Ocean**

Journal:	Royal Society Open Science
Manuscript ID	RSOS-201347
Article Type:	Research
Date Submitted by the Author:	04-Aug-2020
Complete List of Authors:	Schall, Elena; Alfred Wegener Institute Helmholtz Centre for Polar and Marine Research, Ocean Acoustics Lab Thomisch, Karolin; Alfred Wegener Institute Helmholtz Centre for Polar and Marine Research, Ocean Acoustics Lab Boebel, Olaf; Alfred Wegener Institute Helmholtz Centre for Polar and Marine Research, Ocean Acoustics Lab Gerlach, Gabriele; Carl von Ossietzky University of Oldenburg, Animal Biodiversity and Evolutionary Biology; Carl von Ossietzky University of Oldenburg, Helmholtz Institute for Functional Marine Biodiversity Spiesecke, Stefanie; Alfred Wegener Institute Helmholtz Centre for Polar and Marine Research, Ocean Acoustics Lab Van Opzeeland, Ilse; Alfred Wegener Institute Helmholtz Centre for Polar and Marine Research, Ocean Acoustics Lab; Carl von Ossietzky University of Oldenburg, Helmholtz Institute for Functional Marine Biodiversity
Subject:	ecology < BIOLOGY, Oceanography < EARTH SCIENCES, behaviour < BIOLOGY
Keywords:	Passive acoustic monitoring, humpback whale, Megaptera novaeangliae, feeding ground, Southern Ocean, Weddell Sea
Subject Category:	Ecology, Conservation, and Global Change Biology

Author-supplied statements

Relevant information will appear here if provided.

Ethics

Does your article include research that required ethical approval or permits?:

Yes

Statement (if applicable):

Permission for mooring installation were granted to the Alfred Wegener Institute, Helmholtz-Zentrum for Polar- und Meeresforschung by the Federal Environment Office (Umweltbundesamt UBA): Expedition ANT - XXIX/2 UBA permit NÂ° I 3.5 â€“ 94003-3/286.

Data

It is a condition of publication that data, code and materials supporting your paper are made publicly available. Does your paper present new data?:

Yes

Statement (if applicable):

Data on the acoustic presence of humpback whales in the ASSO is uploaded on Dryad (<https://datadryad.org/stash/share/p7jYUrL-GX0It3cDw29k5Ff3HBbhrKNrBxuCO2X7XO0>)

Sea ice concentration data can be downloaded from MathWorks

(<https://www.mathworks.com/matlabcentral/fileexchange/50126-daily-antarctic-sea-ice-concentration>)

Visual sighting data from the ASSO are uploaded on PANGAEA

(<https://doi.org/10.1594/PANGAEA.896842>; <https://doi.org/10.1594/PANGAEA.840382>;

<https://doi.org/10.1594/PANGAEA.819861>; <https://doi.org/10.1594/PANGAEA.819862>;

<https://doi.org/10.1594/PANGAEA.819863>; <https://doi.org/10.1594/PANGAEA.819866>;

<https://doi.org/10.1594/PANGAEA.783806>; <https://doi.org/10.1594/PANGAEA.760340>)

Conflict of interest

I/We declare we have no competing interests

Statement (if applicable):

CUST_STATE_CONFLICT :No data available.

Authors' contributions

This paper has multiple authors and our individual contributions were as below

Statement (if applicable):

[revised manuscript text omitted]

133 monthly starting in January, February, or March.

4. Results

In total, 74,628 hours of recordings were processed, of which 13,049 were *pHWP* hours. Roughly half
of these hours were post-processed by human analysts and, summing all recording locations, 983 hours
were verified as *cHWP* hours (Table 2). Among the four analysts, the level of agreement in classifying
*cHWP* or false positive hours was between 93% and 97%.

*Spatial pattern*

During austral summer and autumn in 2013, nine of the 12 recording positions recorded humpback
whale vocalizations (Table 2). At the positions W10, W11, and W7 humpback whale acoustic presence
could not be confirmed in 2013 (i.e., considering only the even recording hours were included in the
analyses; Table 2, Figure 3). At most recording positions (W9, W8, W6, and W4), the monthly acoustic
presence of humpback whales was not higher than 10% (Figure 3). The recording positions W5 and
W12 registered monthly humpback whale acoustic presences of up to 20% and at the recording position
off Elephant Island (W12), humpback whales were acoustically active during all recorded months of
the year 2013 (Figure 3). The highest monthly acoustic presences of humpback whales (i.e., > 20%)
were confirmed at the four recording positions W3, W2, and W1 on the Greenwich Meridian (Figure
3). Monthly acoustic presences of more than 10% were only registered in areas without sea ice cover
and in most cases monthly acoustic presences of 0% were only registered in areas with a sea ice
concentration of at least 50%. In the central Weddell Sea, humpback whales were only sporadically
acoustically present (i.e., $\leq 10\%$), which presumably was related to the fact that the area was covered
with sea ice over extended periods (Figure 3). At positions W11, W10, and W7, which were covered
by sea ice almost year-round, humpback whales were acoustically absent throughout 2013.

*Intra-annual temporal pattern*

From January until May 2013, at least 50% of recording positions registered humpback whale acoustic
presences (Figure 3, Figure 4). In March 2013, humpback whales were acoustically present at the

largest proportion of the recording positions (9 out of 12). At recording position W6, for example,
humpback whales were acoustically present exclusively during a continuous period of nine days,
between March 18, 2013 and March 21, 2013. At the recording position W5, humpback whales were
acoustically present in January, February, and March 2013 (Figure 4). April 2013 was the month with
the highest proportion of *cHWP* hours summed over all recording positions. Off Elephant Island (W12)
the peak periods for humpback whale acoustic presence were March until May and October through
November (Figure 4). During the months January and February and June until September only sporadic
acoustic presence (i.e., only single hours) was confirmed at the recording position W12 (Figure 4). At
recording positions W8 and W9, similar to Elephant Island, also sporadic acoustic presences of

[revised manuscript text omitted]
*. (ed. ^eds. W. F. Perrin, B. Würsig, J. G. M. Thewissen), pp. 589-592. London: Academic Press.
- Pinto de sa Alves, L. C., Andriolo, A., Zerbin, A., Altmayer Pizzorno, J. L., Clapham, P. 2009 Record of feeding by humpback whales (*Megaptera novaeangliae*) in tropical waters off Brazil. *Publications, Agencies and Staff of the US Department of Commerce*. 45. (doi.org/10.1111/j.1748-7692.2008.00249.x)
- Mikhalev, Y. A. 1997 Humpback whales *Megaptera novaeangliae* in the Arabian Sea. *Marine Ecology Progress Series*. **149**, 13-21. (doi.org/10.3354/meps149013)
- Gibbons, J., Capella, J. J., Valladares, C. 2003 Rediscovery of a humpback whale, *Megaptera novaeangliae*, feeding ground in the Straits of Magellan, Chile. *Journal of Cetacean Research and Management*. **5**, 203-208.
- Stimpert, A. K., Peavey, L. E., Friedlaender, A. S., Nowacek, D. P. 2012 Humpback Whale Song and Foraging Behavior on an Antarctic Feeding Ground. *PLoS ONE*. **7**, e51214. (doi.org/10.1371/journal.pone.0051214)
- Stevick, P. T., Neves, M. C., Johansen, F., Engel, M. H., Allen, J., Marcondes, M. C. C., Carlson, C. 2010 A quarter of a world away: female humpback whale moves 10 000 km between

- breeding areas. *Biology Letters*. **7**, 299-302. (doi.org/10.1098/rsbl.2010.0717)
- Geijer, C. K. A., Notarbartolo di Sciarra, G., Panigada, S. 2016 Mysticete migration revisited: are Mediterranean fin whales an anomaly? *Mammal Review*. **46**, 284-296. (doi.org/10.1111/mam.12069)
- Barendse, J., Best, P. B., Thornton, M., Pomilla, C., Carvalho, I., Rosenbaum, H. C. 2010 Migration redefined? Seasonality, movements and group composition of humpback whales *Megaptera novaeangliae* off the west coast of South Africa. *African Journal of Marine Science*. **32**, 1-22. (doi.org/10.2989/18142321003714203)
- Brown, M. R., Corkeron, P. J., Hale, P. T., Schultz, K. W., Bryden, M. M. 1995 Evidence for a sex-segregated migration in the humpback whale (*Megaptera novaeangliae*). *Proceedings of the Royal Society of London B*. **259**, 229-234. (doi.org/10.1098/rspb.1995.0034)
- Dawbin, W. H. 1966 The seasonal migratory cycle of humpback whales. In *Whales, dolphins and porpoises*. (ed. eds. K. Norris), pp. 145-170. Berkely: University of California Press.
- Thomisch, K., Boebel, O., Clark, C. W., Hagen, W., Spiesecke, S., Zitterbart, D. P., Van Opzeeland, I. 2016 Spatio-temporal patterns in acoustic presence and distribution of Antarctic blue whales *Balaenoptera musculus intermedia* in the Weddell Sea. *Endangered Species Research*. **30**, 239-253. (doi.org/10.3354/esr00739)
- Knox, G. A. 2007 *Biology of the Southern Ocean*. 2nd edition ed: CRC press.
- Nicol, S., Bowie, A., Jarman, S., Lannuzel, D., Meiners, K. M., van der Merwe, P. 2010 Southern Ocean iron fertilization by baleen whales and Antarctic krill. *Fish and Fisheries*. **11**, 203-209. (doi.org/10.1111/j.1467-2979.2010.00356.x)
- International Whaling Commission. 2011 Report on the workshop on the comprehensive assessment of Southern Hemisphere humpback whales. *Journal of Cetacean Research and Management Special Issue* **3**, 1-50.
- Nicol, S., Worby, A., Leaper, R. 2008 Changes in the Antarctic sea ice ecosystem: potential effects on krill and baleen whales. *Marine and Freshwater research*. **59**, 361-382. (doi.org/10.1071/MF07161)
- Deacon, G. 1979 The Weddell Gyre. *Deep Sea Research Part A. Oceanographic Research Papers*. **26**, 981-995. (doi.org/10.1016/0198-0149(79)90044-X)
- Orsi, A. H., Whitworth III, T., Nowlin Jr, W. D. 1995 On the meridional extent and fronts of the Antarctic Circumpolar Current. *Deep Sea Research Part I: Oceanographic Research Papers*. **42**, 641-673. (doi.org/10.1016/0967-0637(95)00021-W)
- Tynan, C. T. 1998 Ecological importance of the Southern Boundary of the Antarctic Circumpolar Current. *Nature*. **392**, 708-710. (doi.org/10.1038/33675)
- Siegel, V. 2016 *Biology and ecology of Antarctic krill*. Springer.
- Van Opzeeland, I., Van Parijs, S., Kindermann, L., Burkhardt, E., Boebel, O. 2013 Calling in the cold: pervasive acoustic presence of humpback whales (*Megaptera novaeangliae*) in Antarctic coastal waters. *PLoS One*. **8**, 1-7. (doi.org/10.1371/journal.pone.0073007)
- Zerbini, A. N., Adriolo, A., Heide-Jørgensen, M. P., Pizzorno, J. L., Maia, Y. G., VanBlaricom, G. R., DeMaster, D. P., Simões-Lopes, P. C., Moreira, S., Bethlem, C., et al. 2006 Satellite-monitored movements of humpback whales *Megaptera novaeangliae* in the Southwest Atlantic Ocean. *MEPS*. **313**, 295-304. (doi.org/10.3354/meps313295)
- Zerbini, A., Andriolo, A., Heide-Jørgensen, M. P., Moreira, S. C., Pizzorno, J. L., Maia, Y. G., Vanblaricom, G. R., Demaster, D. P. 2011 Migration and summer destinations of humpback whales (*Megaptera novaeangliae*) in the western South Atlantic Ocean. *Journal of Cetacean Research and Management, Special Issue*. **3**, 113-118. (doi.org/10.3354/meps313295)
- Rosenbaum, H. C., Maxwell, S. M., Kershaw, F., Mate, B. 2014 Long-Range Movement of Humpback Whales and Their Overlap with Anthropogenic Activity in the South Atlantic Ocean. *Conservation Biology*. **28**, 604-615. (doi.org/10.1111/cobi.12225)
- International Whaling Commission. 2016 Annex H: Report of the Sub-Committee on Other Southern Hemisphere Whale Stocks; 22 May-3 June 2015.
- Teschke, K., Pehlke, H., Deininger, M., Jerosch, K., Brey, T. Scientific background document in support of the development of a CCAMLR MPA in the Weddell Sea (Antarctica)—Version 2016. 2016.
- Deininger, M., Koellner, T., Brey, T., Teschke, K. 2016 Towards mapping and assessing

- Antarctic marine ecosystem services—the Weddell Sea case study. *Ecosystem services*. **22**, 174-192. (doi.org/10.1016/j.ecoser.2016.11.001)
- Dunlop, R. A., Cato, D. H., Noad, M. J. 2008 Non-song acoustic communication in migrating humpback whales (*Megaptera novaeangliae*). *Marine Mammal Science*. **24**, 613-629. (doi.org/10.1111/j.1748-7692.2008.00208.x)
- Payne, R. S., Mcvay, S. 1971 Songs of Humpback Whales. *Science*. **173**, 585-597. (doi.org/10.1126/science.173.3997.585)
- Rettig, S., Boebel, O., Menze, S., Kindermann, L., Thomisch, K., van Opzeeland, I. Year Local to basin scale arrays for passive acoustic monitoring in the Atlantic sector of the Southern Ocean. In: J. Papadakis, L. Bjorno, editors. 1st International Conference and Exhibition on Underwater Acoustics; 2013; Corfu Island, Greece; 2013. p. 1669-1674.
- Baumgartner, M. F., Mussoline, S. E. 2011 A generalized baleen whale call detection and classification system. *The Journal of the Acoustical Society of America*. **129**, 2889-2902. (doi.org/10.1121/1.3562166)
- Klinck, H., Mellinger, D. K., Klinck, K., Hager, J., Kindermann, L., Boebel, O. 2010 Long-range underwater vocalizations of the crabeater seal (*Lobodon carcinophaga*). *The Journal of the Acoustical Society of America*. **128**, 474-479. (doi.org/10.1121/1.3442362)
- Risch, D., Gales, N. J., Gedamke, J., Kindermann, L., Nowacek, D. P., Read, A. J., Siebert, U., Van Opzeeland, I. C., Van Parijs, S. M., Friedlaender, A. S. 2014 Mysterious bio-duck sound attributed to the Antarctic minke whale (*Balaenoptera bonaerensis*). *Biology Letters*. **10**, 20140175. (doi.org/10.1098/rsbl.2014.0175)
- Schall, E., Van Opzeeland, I. 2017 Calls Produced by Ecotype C Killer Whales (*Orcinus orca*) Off the Eckstrom Iceshelf, Antarctica. *Aquatic Mammals*. **43**, 117-126. (doi.org/10.1578/AM.43.2.2017.117)
- Van Opzeeland, I., Van Parijs, S., Bornemann, H., Frickenhaus, S., Kindermann, L., Klinck, H., Plötz, J., Boebel, O. 2010 Acoustic ecology of Antarctic pinnipeds. *Marine Ecology Progress Series*. **414**, 267-291. (doi.org/10.3354/meps08683)
- Stimpert, A. K., Au, W. W., Parks, S. E., Hurst, T., Wiley, D. N. 2011 Common humpback whale (*Megaptera novaeangliae*) sound types for passive acoustic monitoring. *J Acoust Soc Am*. **129**, 476-482. (doi.org/10.1121/1.3504708)
- Cavalieri, D., Parkinson, C., Gloersen, P., Zwally, H. 1996 Sea ice concentrations from Nimbus-7 SMMR and DMSP SSM/I-SSMIS passive microwave data, version 1. *Boulder, Colorado USA, NASA National Snow and Ice Data Center Distributed Active Archive Center*, doi. **10**,
- Greene, C. A. Daily Antarctic sea ice concentration MATLAB Central File Exchange: MATLAB Central File Exchange 2020.
- Greene, C. A., Gwyther, D. E., Blankenship, D. D. 2017 Antarctic Mapping Tools for MATLAB. *Computers & Geosciences*. **104**, 151-157. (doi.org/10.1016/j.cageo.2016.08.003)
- Tynan, C. T., Thiele, D. 2003 Report on Antarctic ice edge definition by the ad hoc working group on ice data collection in the Antarctic. *Paper: SC/55/19, submitted to the Scientific Committee of the International Whaling Commission, paper available from the office of the IWC*. 1p.
- Pomilla, C., Rosenbaum, H. C. 2005 Against the current: an inter-oceanic whale migration event. *Biology Letters*. **1**, 476-479. (doi.org/10.1098/rsbl.2005.0351)
- Garrigue, C., Clapham, P. J., Geyer, Y., Kennedy, A. S., Zerbini, A. N. 2015 Satellite tracking reveals novel migratory patterns and the importance of seamounts for endangered South Pacific humpback whales. *Royal Society Open Science*. **2**, 150489. (doi.org/10.1098/rsos.150489)
- Félix, F., Guzmán, H. M. 2014 Satellite tracking and sighting data analyses of Southeast Pacific humpback whales (*Megaptera novaeangliae*): is the migratory route coastal or oceanic? *Aquatic Mammals*. **40**, 329-340. (doi.org/10.1578/AM.40.4.2014.329)
- Reinke, J., Lemckert, C., Meynecke, J.-O. 2016 Coastal fronts utilized by migrating humpback whales, *Megaptera novaeangliae*, on the Gold Coast, Australia. *Journal of Coastal Research*. **75**, 552-556. (doi.org/10.2112/SI75-111.1)
- Atkinson, A., Siegel, V., Pakhomov, E. A., Rothery, P., Loeb, V., Ross, R. M., Quetin, L. B., Schmidt, K., Fretwell, P., Murphy, E. J., et al. 2008 Oceanic circumpolar habitats of Antarctic krill. *Marine Ecology Progress Series*. **362**, 1-23. (doi.org/10.3354/meps07498)

- Lockyer, C. 1981 Growth and energy budgets of large baleen whales from the southern hemisphere. In *Mammals in the sea vol 3: general papers and large cetaceans*. (ed. ^eds. J. Gordon Clark), pp. 379-487. Rome: Food and Agricultural Organization of the United Nations.
- Owen, K., Kavanagh, A. S., Warren, J. D., Noad, M. J., Donnelly, D., Goldizen, A. W., Dunlop, R. A. 2016 Potential energy gain by whales outside of the Antarctic: prey preferences and consumption rates of migrating humpback whales (*Megaptera novaeangliae*). *Polar Biology*. **40**, 1-13. (doi.org/10.1007/s00300-016-1951-9)
- Cates, K. A., Atkinson, S., Gabriele, C. M., Pack, A. A., Straley, J. M., Yin, S. 2019 Testosterone trends within and across seasons in male humpback whales (*Megaptera novaeangliae*) from Hawaii and Alaska. *Gen. Comp. Endocrinol.* **279**, 164-173. (doi.org/10.1016/j.ygcen.2019.03.013)
- Burkhardt, E. Whale sightings during POLARSTERN cruise ANT-XXVI/3. Alfred Wegener Institute, Helmholtz Centre for Polar and Marine Research, Bremerhaven, PANGAEA 2011.
- Kennedy, A. S., Zerbini, A. N., Vásquez, O. V., Gandilhon, N., Clapham, P. J., Adam, O. 2013 Local and migratory movements of humpback whales (*Megaptera novaeangliae*) satellite-tracked in the North Atlantic Ocean. *Canadian Journal of Zoology*. **92**, 9-18. (doi.org/10.1139/cjz-2013-0161)
- Burkhardt, E. Whale sightings during POLARSTERN cruise ANT-XXVIII/4. Alfred Wegener Institute, Helmholtz Centre for Polar and Marine Research, Bremerhaven, PANGAEA 2013.
- Burkhardt, E. Whale sightings during POLARSTERN cruise ANT-XXVII/2. Alfred Wegener Institute, Helmholtz Centre for Polar and Marine Research, Bremerhaven, PANGAEA 2011.
- Burkhardt, E. Whale sightings during POLARSTERN cruise ANT-XXVII/3. Alfred Wegener Institute, Helmholtz Centre for Polar and Marine Research, Bremerhaven, PANGAEA 2012.
- Burkhardt, E. Whale sightings during POLARSTERN cruise ANT-XXIX/2. Alfred Wegener Institute, Helmholtz Centre for Polar and Marine Research, Bremerhaven, PANGAEA 2013.
- Burkhardt, E. Whale sightings during POLARSTERN cruise ANT-XXVIII/3. Alfred Wegener Institute, Helmholtz Centre for Polar and Marine Research, Bremerhaven, PANGAEA 2013.
- Burkhardt, E. Whale sightings during POLARSTERN cruise ANT-XXVIII/2. Alfred Wegener Institute, Helmholtz Centre for Polar and Marine Research, Bremerhaven, PANGAEA 2013.
- Burkhardt, E. Whale sightings during POLARSTERN cruise ANT-XXIX/3. Alfred Wegener Institute, Helmholtz Centre for Polar and Marine Research, Bremerhaven, PANGAEA 2014.
- Burkhardt, E. Whale sightings during POLARSTERN cruise PS104 (ANT-XXXII/3). Alfred Wegener Institute, Helmholtz Centre for Polar and Marine Research, Bremerhaven, PANGAEA 2018.
- Vu, E. T., Risch, D., Clark, C. W., Gaylord, S., Hatch, L. T., Thompson, M. A., Wiley, D. N., Van Parijs, S. M. 2012 Humpback whale song occurs extensively on feeding grounds in the western North Atlantic Ocean. *Aquatic Biology*. **14**, 175-183. (doi.org/10.3354/ab00390)
- Stevick, P. T., Allen, J., Bérubé, M., Clapham, P. J., Katona, S. K., Larsen, F., Lien, J., Mattila, D. K., Palsbøll, P. J., Robbins, J., *et al.* 2003 Segregation of migration by feeding ground origin in North Atlantic humpback whales (*Megaptera novaeangliae*). *Journal of Zoology*. **259**, 231-237. (doi.org/10.1017/S0952836902003151)
- Stamation, K. A., Croft, D. B., Shaughnessy, P. D., Waples, K. A. 2007 Observations of humpback whales (*Megaptera novaeangliae*) feeding during their southward migration along the coast of southeastern New South Wales, Australia: identification of a possible supplemental feeding ground. *Aquatic Mammals*. **33**, 165. (doi.org/10.1578/AM.33.2.2007.165)
- Morete, M. E., Bisi, T. L., Pace, R. M., Rosso, S. 2008 Fluctuating abundance of humpback whales (*Megaptera novaeangliae*) in a calving ground off coastal Brazil. *Journal of the Marine Biological Association of the UK*. **88**, 1229. (doi.org/10.1017/S0025315408000362)
- Guidino, C., Llapapasca, M. A., Silva, S., Alcorta, B., Pacheco, A. S. 2014 Patterns of Spatial and Temporal Distribution of Humpback Whales at the Southern Limit of the Southeast Pacific Breeding Area. *PLoS ONE*. **9**, e112627. (doi.org/10.1371/journal.pone.0112627)
- Friedlaender, A. S., Tyson, R. B., Stimpert, A. K., Read, A. J., Nowacek, D. P. 2013 Extreme diel variation in the feeding

- behavior of humpback whales along the western Antarctic Peninsula during autumn. *Marine Ecology Progress Series*. **494**, 281-289. (doi.org/10.3354/meps10541)
- Bombosch, A., Zitterbart, D. P., Van Opzeeland, I., Frickenhaus, S., Burkhardt, E., Wisz, M. S., Boebel, O. 2014 Predictive habitat modelling of humpback (*Megaptera novaeangliae*) and Antarctic minke (*Balaenoptera bonaerensis*) whales in the Southern Ocean as a planning tool for seismic surveys. *Deep-Sea Research Part I: Oceanographic Research Papers*. **91**, 101-114. (doi.org/10.1016/j.dsr.2014.05.017)
- Brierley, A. S., Fernandes, P. G., Brandon, M. A., Armstrong, F., Millard, N. W., McPhail, S. D., Stevenson, P., Pebody, M., Perrett, J., Squires, M., *et al.* 2002 Antarctic krill under sea ice: Elevated abundance in a narrow band just south of ice edge. *Science*. **295**, 1890-1892. (doi.org/10.1126/science.1068574)
- Flores, H., van Franeker, J. A., Siegel, V., Haraldsson, M., Strass, V., Meesters, E. H., Bathmann, U., Wolff, W. J. 2012 The Association of Antarctic Krill *Euphausia superba* with the Under-Ice Habitat. *PLoS ONE*. **7**, e31775. (doi.org/10.1371/journal.pone.0031775)
- Lara, R. J., Haas, C., Schnack-Schiel, S. B., Dieckmann, G. S., Kattner, G. 1998 Biological soup within decaying summer sea ice in the Amundsen Sea, Antarctica. *JF Splettstoesser, GAM Dreschhoff (Eds.)*. **73**, 161-171. (doi.org/10.1029/AR073p0161)
- Arrigo, K. R., van Dijken, G. L. 2015 Continued increases in Arctic Ocean primary production. *Progress in Oceanography*. **136**, 60-70. (doi.org/10.1016/j.pocean.2015.05.002)
- Nicol, S. 2006 Krill, currents, and sea ice: *Euphausia superba* and its changing environment. *Bioscience*. **56**, 111-120. (doi.org/10.1641/0006-3568(2006)056[0111:KCASIE]2.0.CO;2)
- Širović, A., Hildebrand, J. A., Wiggins, S. M., McDonald, M. A., Moore, S. E., Thiele, D. 2004 Seasonality of blue and fin whale calls and the influence of sea ice in the Western Antarctic Peninsula. *Deep Sea Research Part II: Topical Studies in Oceanography*. **51**, 2327-2344. (doi.org/10.1016/j.dsr2.2004.08.005)
- Herman, L. M. 2017 The multiple functions of male song within the humpback whale (*Megaptera novaeangliae*) mating system: review, evaluation, and synthesis. *Biological Reviews*. **92**, 1795-1818. (doi.org/10.1111/brv.12309)
- Garland, Ellen C., Goldizen, Anne W., Rekdahl, Melinda L., Constantine, R., Garrigue, C., Hauser, Nan D., Poole, M. M., Robbins, J., Noad, Michael J. 2011 Dynamic Horizontal Cultural Transmission of Humpback Whale Song at the Ocean Basin Scale. *Current Biology*. **21**, 687-691. (doi.org/10.1016/j.cub.2011.03.019)

Tables

Mooring ID	Latitude	Longitude	Recorder ID	Sampling Frequency (Hz)	Deployment Depth (m)
W1 (AWI227)	59 2.82 °S	000 5.78 °E	SV1025	5333	1020
W2 (AWI229)	63 59.85 °S	000 1.84 °E	SV1010	5333	998
W3 (AWI230)	66 2.01 °S	000 3.12 °E	SV1009	5333	949
W4 (AWI232)	68 59.94 °S	000 4.38 °E	SV1011	5333	958
W5 (AWI248)	65 58.09 °S	012 15.12 °W	SV1013	5333	1081
W6 (AWI245)	69 3.480 °S	017 23.32 °W	SV1012	5333	1065
W7 (AWI249)	70 53.55 °S	028 53.47 °W	SV1014	5333	1085
W8 (AWI209)	66 36.45 °S	027 7.26 °W	SV1028	5333	1007
W9 (AWI208)	65 37.23 °S	036 25.32 °W	SV1030	5333	956
W10 (AWI217)	64 22.94 °S	045 52.12 °W	SV1020	5333	960
W11 (AWI207)	63 42.09 °S	050 49.61 °W	SV1033	9600	1012
W12 (AWI251)	61 1.07 °S	055 58.67 °W	SV1008	5333	212

Mooring ID	Total hours	pHWP hours	Hours post-processed	cHWP hours
W1	5,140	584	284	200
W2	5,538	628	306	157
W3	6,316	826	428	203

W4	7,386	1,958	993	23
W5	6,649	1,440	739	77
W6	7,630	163	85	10
W7	6,424	312	157	0
W8	7,077	1,639	823	16
W9	6,973	1,159	594	46
W10	2,767	452	230	0
W11	5,558	923	460	0
W12	7,170	2,965	1,476	251
TOTAL	74,628	13,049	6,575	983

Figures

Figure and table captions

Table 1. Information on passive acoustic recordings included in the dataset. For reference to earlier publications, the original mooring ID is listed in brackets.

Table 2. Overview of recording hours, presumed humpback whale acoustic presence (*pHWP*) hours, post-processed hours, and confirmed humpback whale acoustic presence (*cHWP*) hours per recording location and as an overall sum.

Figure 1. Timeline showing the availability of passive acoustic data collected throughout 2013 for the 12 recording positions in the ASSO.

Figure 2. Bathymetric map of the ASSO and the geographic positions of the 12 bottom-moored recorders included in this study.

Figure 3. Percentage of acoustic presence of humpback whales in the ASSO averaged per recording location and month for the year 2013. Size and color of dots indicates a respective range of percentage of hours per month with humpback whale acoustic presence. The monthly averaged sea ice concentrations are depicted at a 25x25km resolution.

Figure 4. Average proportion of confirmed humpback whale positive (*cHWP*) hours per month at the nine recording positions, W1, W2, W3, W4, W5, W6, W8, W9, and W12 from January until December 2013 (orange bars). Vertical error bars show the respective standard deviations and continuous grey bars represent months without recording data. The blue solid lines and the right y-axis depict the daily averaged sea ice concentration per location within a 50km radius. At three recording positions (W7, W10, and W11) humpback whales were acoustically absent.

Figure 5. Diurnal pattern per month at the recording positions W1, W2, and W12. Proportions of confirmed humpback whale positive (*cHWP*) hours were calculated for each hour of the day and month of the year 2013. The y-axes only display even hours, because only even hours were analyzed in this study.

Appendix B

Response to Editor and Reviewers

Associate Editor Comments to Author

The reviewers have recommended that your paper be accepted following a number of minor revisions. Please ensure that you fully respond to these.

Reply – Thank you very much. We provide answers to all comments in this document.

Reviewer 1 Comments

1. Line 18

"During all months, except December, humpback whale acoustic activity was registered in the ASSO."

The readers should be informed that December was barely sampled. Otherwise, they may misunderstand that there was no humpback whale activity in December. Why was December poorly sampled? How was the logistics to deploy all the moorings? How many cruises? What caused the sampling failures? Logistic problems, electronic malfunctioning, damage due to the ice cover?

Reply – Between the mid of December or the mid of January is usually the time when expedition cruises with Polarstern start and consequently moorings and recorders get installed. The recorders analyzed in this study were installed during the Polarstern cruise ANT-XXIX/2 from December 2012 to February 2013. Depending on the position of the mooring location on the cruise track, recorders get installed earlier or later. Some of the recorders therefore already recorded in December 2012, others started later. The Sonovault recording devices were chosen based on their characteristics for installation at great depths and long recording periods. However, the recorders often do not record the anticipated period and produce time gaps in our data. December is unfortunately the month when recorders often already ran out of battery. We added some additional information in brackets to clarify the issue for the reader (abstract line 11).

2. Lines 89-90, and Table 1 (page 17)

Why was the W11 sampling rate set up to 9,600 Hz while all 11 stations were set up to 5,333 Hz? The authors should provide more information about the hydrophones and setup (sensitivity, digital sample resolution, gain, pre-amp, etc.). Why was the W12 deployed at 212 meters depth, while the others were near 1000 meters? The authors should also provide more information about the mooring line.

Reply – Sampling rates are always set to the highest possible value commensurate with 2 years continuous recordings. As SD card memory capacity increased over years, we were able to record with higher sampling rates. However, due to financial constraints, only few recorders were upgraded with larger SD cards. We added additional information on the hydrophones and setup in line 69-70. Hydrophones are deployed piggy-back to HAFOS moorings, which in deep waters typically extended to 800m as the shallowest depth (to avoid damage by drifting ice bergs). Hydrophones were mounted 200m below the top floatation, resulting in 1000m recording depth. The water depth at the W12 mooring off Elephant Island is only 300m, restricting deployment depth at this site. We added additional

information on the mooring set up in line 72-82 and included an additional reference where detailed information on the HAFOS mooring set up can be found.

3. Lines 105-107

What is the impact of analyzing only even hours? Is the number of cHWP underestimated? What does it mean? Do cHWP in consecutive even hours may indicate cHWP in intermediate odd hours?

Reply – Comparative analyses were carried out to evaluate exactly this. We have added a section in line 99-107, explaining this in more detail.

4. Lines 130-133: "In order to test for correlations between humpback whale acoustic presence and the local sea ice concentration, the Pearson correlation coefficient was calculated for 4 different temporal regimes: Monthly and three-monthly starting in January, February, or March."

I suggest that the authors rephrase the sentence in order to clarify this topic.

Reply – The sentence was rephrased in line 146-149.

5. Lines 141-142: "During austral summer and autumn in 2013, nine of the 12 recording positions recorded humpback whale vocalizations (Table 2)."

It should be noted that the hydrophone W10 was deployed after summer, and the majority of the hydrophones did not fully sample the summer season. I suggest annotating the months in parentheses after "summer and autumn.". In Figure 1, I also suggest drawing vertical lines to separate the seasons.

Reply – The suggested annotation of months was included in line 157 and vertical lines were added in Figure 2 (Figure 1 before) to indicate seasons.

6. Lines 154-155: "At positions W11, W10, and W7, which were covered by sea ice almost year-round, humpback whales were acoustically absent throughout 2013."

The information is related to the lines 141-142. They could be merged.

Reply – The concerning sentence deals with the acoustic absence due to probably too high sea ice concentrations, which we believe fits best at the end of the paragraph when also sea ice concentrations at the other positions are mentioned.

7. Lines 169-172: "At the recording positions W1 through W3 at the Greenwich Meridian the acoustic presence of humpback whales was strongly seasonal: humpback whales were acoustically present between January and July with peak periods in March until June (depending on the position; Figure 4)."

I am not sure if it is wise to affirm that considering the lack of sampling—the W1 station stops in July, W2 in August, and W3 in late September.

Reply – We understand the point of the reviewer; however, we still believe that there is a clear sign of seasonality: Humpback whale vocalizations disappear from the recordings at W2 and W3 in May although the recorders kept recording for at least two more months. Recordings at W1 in July only contained very little humpback whale vocalizations indicating

that most of the individuals have left the area during this time. From preliminary analyses of data from 2011, 2012, 2017, and 2018 from the same recording positions, we additionally detected the same seasonal trend.

8. Line 215: "continents."

Reply – The typo was adjusted in line 247.

9. Line 227: "observed."

Reply – The typo was adjusted in line 259.

10. Line 238: "70°S."

Reply – The typo was adjusted in line 270.

11. Lines 250-254: "Summarizing all recording positions, our data indicate humpback whale presence in the ASSO during all months of the year, except December. December could either be the month with the fewest or no humpback whales present in the ASSO, or represent a period during which whales do not or only rarely vocalize. However, for all locations, overall data coverage for December was poor (only a few days during December 2012) which could have affected detection probability of calls."

Considering data coverage for December was poor, as noted by the authors, the phrase "December could either be the month with the fewest or no humpback whales present in the ASSO, or represent a period during which whales do not or only rarely vocalize." turns into speculation. It is not possible to make conclusions based on a lack of observation due to poor sampling.

Reply – We understand the point of the reviewer and adjusted the content of the paragraph accordingly. The speculative part is still also plausible for the months January and February, when data coverage is good. Therefore, we moved the possible explanation down in the paragraph and adjusted its content so it explains the low acoustic activity during summer in general rather than only one month which was poorly sampled (see line 297-299).

12. Line 257: "humpback whales are regularly sighted in the ASSO..."

I suggest adding the references:

DALLA ROSA, L. ; SECCHI, E. R. ; KINAS, P. G. ; SANTOS, M. C. O. ; MARTINS, M. B. ; ZERBINI, A. N. ; BETHLEM, C. . Photo-identification of humpback whales, *Megaptera novaeangliae*, off the Antarctic Peninsula: 1997/98 to 1999/2000. *Memoirs of the Queensland Museum*, v. 47, n.2, p. 555-561, 2001.

SECCHI, E. R.; DALLAROSA, L. ; KINAS, P. G. ; SANTOS, M. C. O. ; ZERBINI, A. N. ; BASSOI, M. ; MORENO, I. B. . Encounter rates of whales around the Antarctic Peninsula with special reference to humpback whales, *Megaptera novaeangliae*, in the Gerlache Strait: 1997/98 to 1999/2000. *Memoirs of the Queensland Museum*, v. 47, n.2, p. 571-578, 2001.

Reply – We included the reference Dalla Rosa et al. (2001) in line 300. The reference Secchi et al. (2001) was not included because the sighting data presented in this study stems

exclusively from the Pacific side of the Antarctic Peninsula. Therefore, this second reference is not supporting the point of regular humpback whale sightings in the ASSO.

13. Line 265: "the waters around the Elephant Island..."
Please, consider to include a legend on the map (Figure 2).

Reply – Labels for important locations were included in the map in Figure 1 (before Figure 2).

14. Lines 274-276: "A high proportion of the acoustic activity during these months was attributed to singing humpback whale males (Schall et al. unpublished data).
"attributed", instead of "atributed."
Please, provide more information about the reference Schall et al. since it is used to support an important statement.

Reply – Typo was adjusted in line 318. 'Schall et al. unpublished data' refers to an ongoing study on humpback whale song in the ASSO. We added some additional information in line 319 in brackets.

15. Figure 4
Please, include panels for W7, W10, and W11.

Reply – The graphs for W7, W10, and W11 were left out on purpose because no acoustic presence was detected during no times at these locations (see line 158-159 in the results section). The graphs would be empty and would therefore only take up space without showing any information. We have added a sentence to the caption to clarify this.

16. Figure and table captions. Figure 4.
"Average proportion of confirmed humpback whale presence (cHWP)..."

Reply – The typo was adjusted in the captions of Figure 4 and 5.

Reviewer 2 Comments

1. Line 13 - "allows to study" consider changing to "allows the study of"

Reply – The sentence was adjusted in line 6 of the abstract.

2. Line 35 - "extent" to "extend"

Reply – The typo was adjusted in line 13.

3. Lines 51-52 - an isolating current - see comment on pdf

Reply – 'isolating' was changed to 'insulating' in line 30 because this suggested adjective also describes the function of the current well.

4. Lines 64-66 - Revise for clarity

Reply – The sentence was rephrased for clarity (see line 44-46).

5. Line 88 - Consider revising figure names

Reply – The figure order was changed accordingly in line 68 as well as in the figure and figure caption sections.

6. Lines 132-133- Revise for clarity

Reply – The sentence was rephrased for clarity in line 146-149.

7. Lines 147-148 - Revise for clarity - Including pHWP? Because according to cHWP, no humpback whales were recorded at any recorders.

Reply – pHWP hours were not included in any of the results, neither in graphics nor in numbers in the text. Only cHWP hours were considered in the results and interpreted in the discussion (see line 152-155).

8. Lines 166-169 - Revise for clarity

Reply – Sentence was rephrased for clarity (see line 191-192).

9. Lines 177-179 - I would specify the reason that W3 - W11 were not included in Figure 4. At this point it is only intuitive after reading the next sentence. Additionally you say that W2 show random HW diurnal presence yet still include it in Figure 4 which may be considered slightly confusing with regards to which recorders were included and which not.

Reply – We corrected the typo of having listed W2 for both types of diurnal patterns in line 202. We also added the additional information that W7, W10, and W11 are not mentioned due to the fact that no vocalizations were recorded at all at these positions in line 210-211 and we included an additional explanation in the figure caption of Figure 4.

10. Lines 270-271 - Revise for clarity

Reply – We rephrased the sentence for clarity (see line 313-314).

11. Line 365 - Check reference format inconsistencies

Reply – References were checked for inconsistencies.